# Fibroblast Activation Protein Is Expressed by Altered Osteoprogenitors and Associated to Disease Burden in Fibrous Dysplasia

**DOI:** 10.3390/cells13171434

**Published:** 2024-08-27

**Authors:** Layne N. Raborn, Zachary Michel, Michael T. Collins, Alison M. Boyce, Luis F. de Castro

**Affiliations:** 1Skeletal Disorders and Mineral Homeostasis Section, National Institute of Dental and Craniofacial Research, National Institutes of Health, Bethesda, MD 20892, USA; laynenr123@gmail.com (L.N.R.); mcollins@nih.gov (M.T.C.); 2Metabolic Bone Disorders Unit, National Institute of Dental and Craniofacial Research, National Institutes of Health, Bethesda, MD 20892, USA; zachary.michel@nih.gov (Z.M.); alison.boyce@nih.gov (A.M.B.)

**Keywords:** fibrous dysplasia, fibroblast activation protein, biomarker, protease

## Abstract

Fibrous dysplasia (FD) is a mosaic skeletal disorder involving the development of benign, expansile fibro-osseous lesions during childhood that cause deformity, fractures, pain, and disability. There are no well-established treatments for FD. Fibroblast activation protein (FAPα) is a serine protease expressed in pathological fibrotic tissues that has promising clinical applications as a biomarker and local pro-drug activator in several pathological conditions. In this study, we explored the expression of *FAP* in FD tissue and cells through published genetic expression datasets and measured circulating FAPα in plasma samples from patients with FD and healthy donors. We found that *FAP* genetic expression was increased in FD tissue and cells, and present at higher concentrations in plasma from patients with FD compared to healthy donors. Moreover, FAPα levels were correlated with skeletal disease burden in patients with FD. These findings support further investigation of FAPα as a potential imaging and/or biomarker of FD, as well as a pro-drug activator specific to FD tissue.

## 1. Introduction

Fibrous dysplasia (FD) is a mosaic skeletal disorder arising from acquired activating variants in *GNAS* in skeletal stem cells during embryogenesis. When extra-skeletal tissues are also affected, it is called McCune-Albright Syndrome (MAS, OMIM #174800). FD results in the development of benign expansile fibro-osseous lesions in the skeleton during childhood, leading to deformity, fractures, pain, and disability. As with most mosaic disorders, the disease burden is highly variable, ranging from one single lesion (monostotic FD) to the involvement of large portions of the skeleton (polyostotic FD). The histopathology of FD lesions is characterized by the replacement of normal bone and marrow tissue by fibrous tissue mixed with curvilinear trabeculae of poorly mineralized and hypercellular woven bone. FD fibroblasts are highly proliferative abnormal bone marrow stromal cells (BMSCs). These cellular changes are caused by the expression of hyper-signaling Gα_s_, which produces increased intracellular cyclic adenosine monophosphate (cAMP), leading to the abnormal differentiation and increased proliferation of BMSCs.

There are no well-established treatments for FD. FD BMSCs produce large amounts of RANKL, promoting the local differentiation and activation of osteoclasts, which in turn contribute to FD BMSC proliferation in a positive feedback loop [1]. Treatment with the anti-RANKL monoclonal antibody denosumab, which dramatically inhibits osteoclastogenesis, has shown promising therapeutic effects in a clinical trial [2]. But denosumab does not discriminate between affected and unaffected skeletal tissue, decreasing osteoclastogenesis in the complete skeleton. Osteoclasts are necessary for bone homeostasis, so in principle this therapy should eventually be discontinued to allow normal bone modelling and remodeling to resume. This is especially true for children, who are also the main beneficiaries of FD therapy, but need osteoclasts for appropriate skeletal development. Denosumab discontinuation leads to disease rebound and can cause severe hypercalcemia due to the rapid reactivation of osteoclastogenesis [2,3]. In addition to denosumab, small molecules that specifically target activated Gα_s_ variants are being investigated, but most of the molecules identified also have inhibitory action towards wildtype Gα_s_ [4]. Since FD is a mosaic disease, treatments that specifically target lesional tissue would greatly improve these and other possible treatment drawbacks. Research that focuses on the development of pro-drugs which are activated only within FD lesions may be an answer.

Fibroblast activation protein (FAPα) is a membrane-bound serine protease involved in extracellular matrix degradation. It was originally identified within the stroma of solid tumors, expressed by carcinoma-associated fibroblasts [5]. It can be naturally cleaved and released into the circulation, and since its expression is relatively restricted to fibrotic pathological processes, it has the potential to be utilized as both a circulating biomarker and therapeutic target in an array of diseases involving fibrosis and inflammation [6]. Small molecules and monoclonal antibodies capable of targeting and inhibiting FAPα were developed for cancer treatment. However, they failed to significantly improve disease progression in clinical trials to justify further research on their use as antitumor agents [7,8,9]. Nevertheless, more recent literature has shown that radiolabeled FAPα inhibitors (^68^Ga-FAPI and ^18^F-FAPI) can be effective when used as medical imaging pan-tumoral radiotracers in solid tumors [10,11]. FAPα is also a promising pro-drug activating agent for engineered drugs [12,13,14] as it has both proteolytic cleavage site specificity [15], and expression that is relatively confined to pathological fibrotic tissue.

This work follows our recent publication that screened released factors by FD BMSCs that may have therapeutic or diagnostic interest [16]. Here, we report that patients with FD have increased blood FAPα levels compared to healthy donors (HDs), and that there is a clear association of these levels with their disease burden. In addition, we collected and integrated *FAP* gene expression levels reported in previous publications and publicly available FD-related mRNA expression data published by us, and others. In this search, we found increased *FAP* mRNA levels in murine FD tissue and cells, as well as in patient-derived FD cells when compared to HDs. We did not observe significant expression changes in human BMSCs engineered to express Gα_s_^R201C^. Lastly, we report that anti-RANKL therapy decreased *FAP* expression in human and murine FD tissue.

## 2. Material and Methods

### 2.1. Literature Search

Independent searches and literature reviews were conducted by LNR, ZM and LFdC between 6th August and 9th 2024, following the workflow in Figure 1, which resulted in 10 publications reviewed for *FAP* expression (Table 1). The full list of articles identified by these searches is included as Appendix A.

### 2.2. Human Specimens

Forty-seven adult patients with FD were evaluated at the NIH Clinical Center as part of a longstanding natural history protocol (NIH 98-D-0145, NCT00001727 in www.clinicaltrials.gov). The study was approved by the NIDCR Institutional Review Board and all subjects gave informed consent/assent. FD disease burden was calculated using a validated method (Skeletal Burden Score [SBS]) [16]. This study includes patients undergoing treatment with bisphosphonates, and those who paused therapy for more than a year before donating blood for this study, who were considered off-drug (Table 2). Serum samples from 22 HDs were purchased (Valley Biomedical, Winchester, VA, USA) and donors were considered to have SBS = 0. The demographic information of both subject groups is reported in Table 2 and individual values are available in Appendix A. 

### 2.3. FAPα Determination

An FAPα Human ELISA Kit with a sensitivity of 12 pg/mL and range of 12–4000 pg/mL was purchased from Abcam (Cambridge, UK, catalog number ab193701) and used to measure FAPα in plasma samples according to the manufacturer’s instructions. Plasma samples were diluted 200-fold prior to ELISA assay as recommended by the manufacturer. 

### 2.4. FAP/Fap mRNA Expression Level Analyses

Expression data from publicly available datasets was obtained from NCBI GEO repository (Table 1) and differential expression of *FAP* between sample groups was calculated. The methods for the obtention and characteristics of these samples are described in their corresponding publications.

### 2.5. Statistical Analysis

Statistical analysis of data in Figure 2 is described in their respective publications. For plasmatic FAPα levels and SBS data we used GraphPad Prism 9.00 (GraphPad Software, Inc., La Jolla, CA, USA), and conducted group comparisons using unpaired, two-tailed *t*-test after evaluating normality with the D’Agostino-Pearson test. Spearman correlation was calculated for plasmatic FAPα levels and SBS. For these analyses, SBS in HDs was considered 0. A linear regression of plasmatic FAPα determinations and SBS was also performed.

## 3. Results

The literature search for high throughput mRNA expression analyses of FD tissue and cultured FD cells resulted in 10 publications. Of these, five were excluded, as their analysis did not include FAP. One study did not report *FAP* as a gene modulated in any of the pathologies assessed and did not provide a list of genes measured [24]. One study included *FAP* expression in single-cell RNAseq from three FD tissue samples, but did not report differential expression in comparison to normal cells, and thus was excluded [17] (Figure 1, Table 1). Three publications measured *FAP* expression as part of high throughput mRNA datasets. One of these three studies is our previously reported article in which we also demonstrated increased secretion of FAPα protein into the culture media of BMSCs derived from a doxycycline-inducible FD mouse model [16]. This data has been reproduced again in Figure 2A. In this study, bone marrow was plated, cleared from hematopoietic cells by negative immunoselection of CD45+ cells to enrich BMSCs, and split in 6-well plates. Some wells were induced in vitro to express Gα_s_^R201C^ by adding doxycycline in the media, and media levels of FAPα were measured, showing a 4.4-fold increase in Gα_s_^R201C^-expressing cultures compared to uninduced cultures. However, no significant differences were found at the mRNA expression level in these cultures 48 h after initiating Gα_s_^R201C^ expression induction. In the same study, using human cells, media FAPα levels did not show a significant difference between FD and HD BMSCs. However, measurements of additional well-known disease-associated factors in these samples also failed to show significant differences to control cultures, likely due to the high variability involved in the study of human samples from multiple donors, as discussed in the publication [16]. However, these human cultures showed a significant increase in *FAP* mRNA expression (Figure 2B). On the other hand, analyses from an additional mRNA expression dataset of human BMSCs engineered by lentiviral transduction to express Gα_s_^R201C^ showed no changes in the expression levels of *FAP* [19]. 

In a different study we published [1,25], FD lesions were induced by the conditional expression of Gα_s_^R201C^ in the appendicular skeleton of mice. RNAseq relative expression analysis of the lesions in comparison to normal bone also showed a significant increase in *Fap* expression, which was partially normalized after four weeks of anti-RANKL therapy (Figure 2C). In the same publication, human FD biopsies from a Phase 1 clinical trial obtained before and after denosumab (anti-RANKL monoclonal antibody) treatment showed reduced expression of *FAP* after six months of denosumab therapy (Figure 2D).

With this preliminary evidence, we explored the role of FAPα as a biochemical biomarker of FD, comparing its blood levels in 47 patients with FD to that of 22 healthy donors (Table 1). FD patients showed plasmatic FAPα levels more than twice as high as healthy donors (Figure 3A), which was strongly correlated with FD disease burden (Figure 3B).

## 4. Discussion

The evidence reported here demonstrates that *FAP* has increased expression in FD tissue, and more specifically, in abnormal FD BMSCs. Given its expression as a defining characteristic of pathological fibrosis, its potential as a pro-drug cleaving activator, and the availability of FAPα-targeting drugs with good tolerability in humans [7,8], FAPα has attracted the attention of researchers across disciplines studying disorders involving fibrosis. This positions FAPα as an interesting translational research candidate in FD, with potential to improve diagnosis and treatment. 

FAPα may be useful in the imaging realm of disease workup, however, additional studies are needed to show its performance compared to the current standard. Radiolabeled FAPα inhibitors (^68^Ga-FAPI and ^18^F-FAPI) imaging by PET/CT have shown efficacy in labeling a wide range of solid carcinomas via interactions with cancer-associated fibroblasts [10,11], and may improve lesional localization and activity measurement in FD. Two bone-avid molecules, 99m-technetium-methylene diphosphonate (^99^mTc-MDP) and more recently ^18^F-NaF [25] effectively demarcate skeletal FD lesions and are essential in determining disease burden and guiding patient treatment plans. In addition, ^18^F-NaF (and not ^99^mTc-MDP) accurately captures lesion activity through analysis of local standard uptake values (SUV) which can be used as a measurement of lesion improvement [2]. However, considering that their mechanism of action involves integration with mineralized tissue, it is unclear if these radiotracers have the capacity to target FD lesions with low or absent mineral content. In contrast, radiolabeled FAPα inhibitors may offer visualization for non-skeletal lesions in FD/MAS, i.e., intramuscular myxomas, or others that remain to be characterized [26]. Indeed, ^18^F-FAPI PET/CT has already been tested on a patient with FD [27] and it showed increased uptake in lesional tissue compared to ^18^F-Fludeoxyglucose, which is a poor radiotracer for FD. Comparative studies imaging the same patients using ^18^F-FAPI and ^18^F-NaF PET/CT, the current standard, are necessary to better understand the way these radiotracers perform in FD.

FAPα may also be useful as a biomarker. We demonstrated a correlation of FAPα circulating levels and disease burden that is as strong as other well-known FD biomarkers such as RANKL, the bone turnover markers alkaline phosphatase (ALP), osteocalcin, C-terminal telopeptide of type I collagen (CTX-I), and N-terminal pro-peptide of type I procollagen (PINP) [16]. Importantly, while bone turnover markers are associated with virtually all skeletal pathologies, FAPα is highly specific for fibrotic tissue. Including FAPα with bone turnover markers in FD diagnostic blood panels could provide both diagnostic and prognostic insights for FD. Studies of FAPα in additional cohorts of FD patients may confirm our observations.

FAPα has a relatively specific cleavage substrate that can be engineered as a pro-drug lock, targeting activation in *FAP*-expressing diseased tissue. Pharmacologic research has recently focused on tethering drugs to locking peptides, which allows their local activation in target tissues [28]. This can be particularly beneficial in mosaic diseases which involve variable regions of the body, rather than full organs or systems. FD’s unique fibro-osseous histology shows production and rapid remodeling of large amounts of extracellular matrix in which *FAP* is distinctively expressed. In addition, FAPα has a relatively specific substrate cleavage site, allowing it to be recognized and cleaved by engineered drugs. Several proof of principle study assays have been carried out to explore the capacity of FAPα as a local pro-drug activator of engineered cytotoxic pro-drugs for cancer treatment, including the combination with photoactivation in peptide-locked photosensitizers [12,13,14]. In addition, MMP2 is another protease highly expressed in FD BMSCs [16,19,25,29]. However, it is more broadly expressed in non-pathological tissues and its target cleavage site overlaps with those of other metalloproteases. Nevertheless, an anti-TNFα antibody locked with a peptide cleavable by MMP2/9 showed therapeutic effects equivalent to unmodified anti-TNFα in a mouse model of rheumatoid arthritis, but animals treated with MMP2/9 activable pro-drug lacked the systemic immunodepression secondary effects that animals receiving the unmodified antibody showed [29]. Denosumab is an anti-osteoclastic, RANKL-neutralizing antibody with demonstrated efficacy for FD treatment. However, it leads to a systemic abrogation of osteoclasts, which are necessary for bone growth and turnover. This obliges treatment discontinuation, which in turn generates a rapid rebound of resorption activity and associated co-morbidities such as acute hypercalcemia and FD lesion reactivation. With these considerations, creating a denosumab-like monoclonal antibody with a FAPα-cleavable peptide that only activates in *FAP*-expressing FD lesions could allow sustained therapy by eliminating off-target effects and potentially promote local reactivation of normal bone turnover as lesions normalize their cellular composition and reduce *FAP* expression.

## 5. Conclusions

FAPα is a serine protease specifically expressed in FD that has potential to be used as a disease biomarker in blood biochemical tests and medical imaging. Its relatively specific target cleavage sequence makes it a promising candidate as a local pro-drug activator in FD lesions. This exploratory study may open novel translational research avenues to test the clinical applications enabled by FAPα in FD.

## Figures and Tables

**Figure 1 cells-13-01434-f001:**
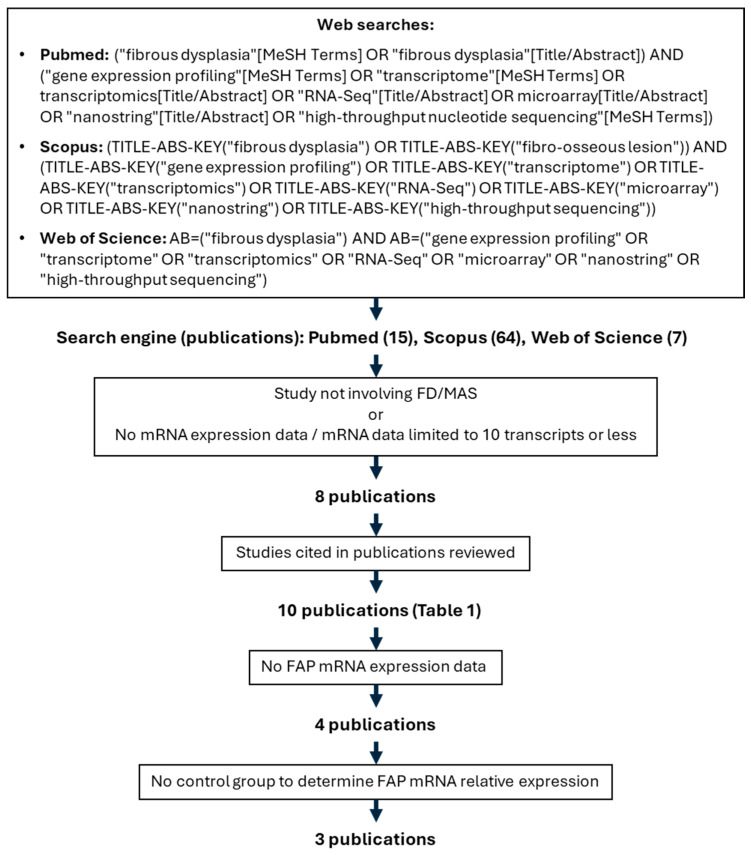
Flow chart showing search terms and inclusion criteria for *FAP* mRNA relative expression analysis.

**Figure 2 cells-13-01434-f002:**
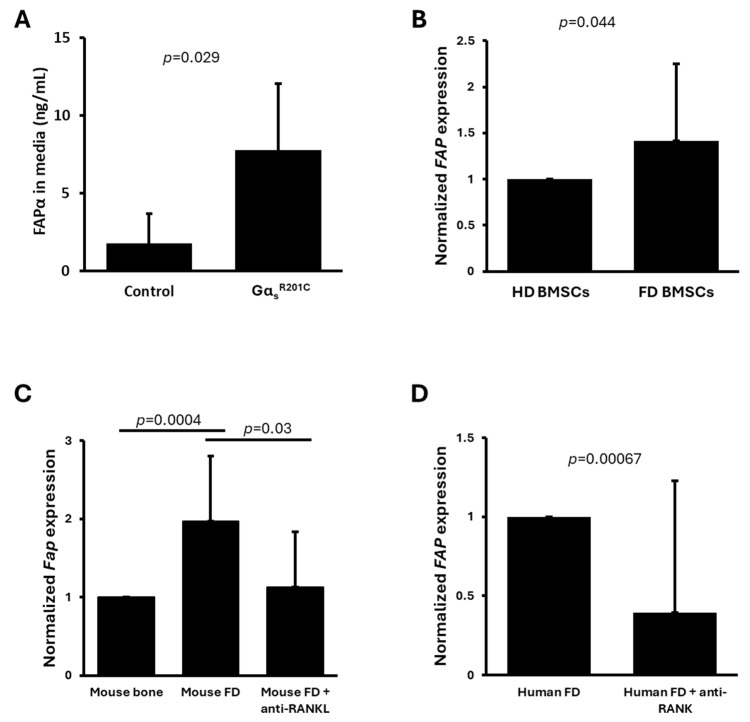
FAPα is produced by altered bone marrow stromal cells (BMSCs) in fibrous dysplasia (FD) tissue, and its expression is normalized with anti-RANKL therapy. Data retrieved and adapted from published studies and genetic expression datasets [1,16]. (**A**) FAPα is secreted by mouse BMSCs cultures induced to express FD-causing Gα_s_^R201C^ (*n* = 5/group) [16]. (**B**) *FAP* expression is upregulated in patient-derived FD BMSC compared to healthy volunteers (HD) BMSCs (*n* = 4/group) [16]. (**C**) *Fap* expression is upregulated in mouse FD lesions compared to healthy littermate bone tissue, and anti-RANKL therapy normalizes its expression (*n* = 6 WT mice, 5 FD mice, 6 anti-RANKL-treated FD mice) [25]. (**D**) *FAP* expression is downregulated in human FD tissue after anti-RANKL therapy with denosumab (*n* = 6) [25]. Data is presented as averages and standard deviation in (**A**) and average and standard error in (**B**,**C**).

**Figure 3 cells-13-01434-f003:**
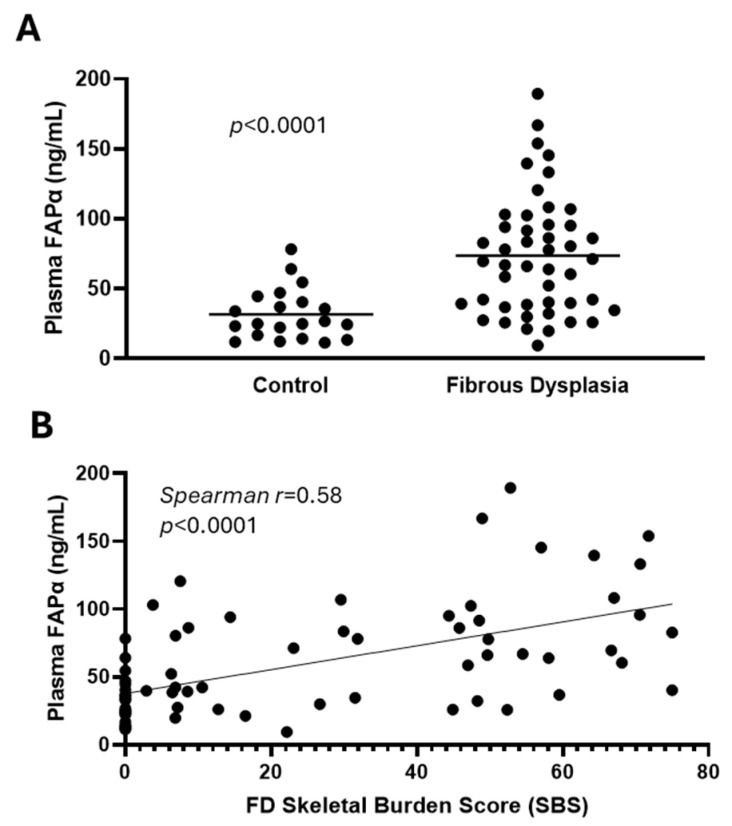
Circulating FAPα concentration is significantly higher in plasma from fibrous dysplasia (FD) patients than from the plasma of healthy donors (HD) and is associated with their disease burden. (**A**) Level of FAPα in plasma from FD patients compared to HD. (**B**) Correlation of FAPα with skeletal disease burden score of FD patients. HDs were considered to have a score of zero (absence of disease). The line represents linear regression.

**Table 1 cells-13-01434-t001:** Datasets reviewed for *FAP* expression.

Study [Ref]	Sample Type	GEO Accession	mRNA Expression Technique	*FAP* Expression Measured?	*FAP* Upregulation
Kim, HY, 2024 [17]	Single cells from FD tissue	GSE263294	scRNAseq	Yes (scRNAseq)	N/A, no control group
Kim, HY, 2024 [17]	FD BSMC vs. normal BMSC-derived organoids (culture)	Not submitted	qPCR	No	N/A
Michel Z, 2024 [16]	Human FD and HD BMSCs	GSE261360	RNAseq	Yes	Yes, FD vs. HD BMSCs
Michel Z, 2024 [16]	Mouse BMSCs induced or not to express Gα_s_^R201C^	GSE261360	RNAseq	Yes	No
De Castro LF, 2023 [1]	Human FD tissue before and after denosumab treatment	GSE250357	RNAseq	Yes	Yes, FD vs. denosumab-treated FD
De Castro LF, 2023 [1]	Mouse FD tissue with or without anti-RANKL treatment and same site control bone	GSE250357	RNAseq	Yes	Yes, FD vs. anti-RANKL-treated FD and vs. control bone
Persichetti, 2021 [18]	FD biopsies	GSE176243	Nanostring	No	N/A
Raimondo D, 2020 [19]	Human BMSC expressing Gα_s_^R201C^ by lentiviral transduction	GSE109818	Microarray	Yes	No
Onodera S, 2020 [20]	GNAS p.R201H iPSC vs. WT iPSC	Not submitted	qPCR	No	N/A
Zhou S-H, 2014 [21]	Human FD tissue vs. normal bone	Not submitted	Microarray	No	N/A
Piersanti S, 2010 [22]	Human BMSC expressing Gα_s_^R201C^ by lentiviral transduction	Not submitted	qPCR	No	N/A
Kiss J, 2010 [23]	FD affected women’s tissue vs. control women’s bone	Not submitted	qPCR	No	N/A
Lee C-H, 2008 [24]	FD biopsies vs. GCT and ABC biopsies	Not submitted	Microarray	Unk *	Unk

Abbreviations: FD = Fibrous dysplasia, N/A = Not applicable, BMSC = Bone marrow stromal cells, HD = Healthy donor, *FAP* = Fibroblast activation protein, scRNAseq = Single cell RNA sequencing, RANKL = Receptor activator of nuclear factor kappa-Β ligand, iPSC = Induced pluripotent stem cell, GCT = Giant cell tumor, ABC = Aneurysmal bone cyst, Unk = Unknown. * Assessed gene list not available in the publication or its references, *FAP* was not reported as a differentially expressed gene in any of the sample groups assessed.

**Table 2 cells-13-01434-t002:** Demographic data of plasma donors.

	Fibrous Dysplasia (FD) Patients	Healthy Donors (HDs)
Subjects—*n*	47	22
Females—*n* (%)	30 (63.8%)	8 (36.4%)
Age—mean ± SD (range)	33 ± 13 (18–71)	42 ± 15 (18–70)
FD burden—mean ± SD (range)	37 ± 23 (2.8–75)	0
Currently on bisphosphonates—*n* (%)	10 (21%)	0
Any endocrinopathy—*n* (%)	35 (74%)	0

## Data Availability

Literature search results are available in Appendix A. GEO gene expression datasets reviewed on 6 to 9 August 2024 for the publications in Table 1 are indicated in the GEO accession column and can be accessed at https://www.ncbi.nlm.nih.gov/geo/, accessed on 9 August 2024. Data reproduced in Figure 2 has been adapted from previous publications [1,16]. Data in Figure 3 and Table 2 is available in Appendix A.

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
