# Peer review of "Fibroblast Activation Protein Is Expressed by Altered Osteoprogenitors and Associated to Disease Burden in Fibrous Dysplasia"

_cells, 2024, doi:10.3390/cells13171434_

Round 1

Reviewer 1 Report

Comments and Suggestions for Authors

Dear Authors,

Thank you very much for submitting your research to Cells

1. Why did the authors only use Pubmed?

2. Who participated in the selection process?

3. Please draw flow chart for the selection process.

4. What is the main message the authors would like to deliver to the readers?

5. Authors have submitted the manuscript as Brief Report. However, the results needs to be explained in more detailed fashion.

6. What is the true novelty of this manuscript.

7. Authors have mentioned about the funding. What is the connection between this research and the funding?

8. Please provide the future research regarding the topic.

9. What is the clinical relevance to this study.

Thank you very much.

Author Response

We appreciate your review of our manuscript which contributed to increase the quality of our submission. We hope we clarified your questions.

  1. Why did the authors only use Pubmed?

We chose PubMed for our literature review because it is a comprehensive and widely used database for biomedical literature. However, to improve the quality of the selection process, we now included searches on two additional scientific search engines: Scopus and Web of Science. Scopus search identified an additional publication with high throughput mRNA expression data on fibrous dysplasia, although FAP was not included in the list of genes reported. This publication was added to table 1. We also submitted the full list of publications identified as a supplementary spreadsheet.

  1. Who participated in the selection process?

Layne N Raborn, Zachary Michel and Luis F de Castro carried out these searches and each reviewed all the publications independently. This is now reflected in Materials and Methods (line 70)

  1. Please draw flow chart for the selection process.

We included a flow chart for the selection process as figure 1 (line 201)

  1. What is the main message the authors would like to deliver to the readers?

The main message of our study is that Fibroblast Activation Protein alpha (FAPα) is significantly overexpressed in fibrous dysplasia (FD) tissues and cells, and its levels in plasma correlate with disease burden. This suggests that FAPα could serve as a valuable biomarker for FD and potentially as a target for pro-drug activation within FD lesions.

  1. Authors have submitted the manuscript as Brief Report. However, the results needs to be explained in more detailed fashion.

We edited the results section of the manuscript to better explain the literature search and selection process and to further clarify that the data reported in figure 2 was retrieved from previously published articles. mRNA relative expression data (Fig 2 B-D) was included in GEO datasets submitted for publications #15 and #19 that have now been re-analyzed and integrated to explore the role of FAP in FD.

  1. What is the true novelty of this manuscript.

In addition to searching and integrating data regarding the expression status of FAP in FD from publicly available gene expression datasets, we demonstrated for the first time that FAPα is significantly elevated in FD patients' plasma and correlates with disease burden. This finding opens new avenues for using FAPα as a diagnostic biomarker and for the development of targeted therapies specifically for FD.

  1. Authors have mentioned about the funding. What is the connection between this research and the funding?

Our research was funded by the Intramural Research Program at the National Institute of Dental and Craniofacial Research (NIDCR), which provided support for our investigation into FD. The funding facilitated the acquisition of essential resources, equipment, and specimen samples necessary for our study, thereby enabling us to explore new diagnostic and therapeutic strategies for FD. Fund obtained from Foundation for the NIH from the Doris Duke Charitable Foundation, Genentech, the American Association for Dental Research, the Colgate-Palmolive Company were used to support the fellowship stipend of one of the authors, as part of the NIH Medical Research Scholars Program.

  1. Please provide the future research regarding the topic.
  2. What is the clinical relevance to this study.

Please, find below a combined answer for comments 8 and 9. There are three research directions closely associated with the clinical relevance of FAPα that may be further explored by us or others in the future.

  • Plasmatic levels of FAPα can be used as a biomarker of FD disease. This is discussed in the manuscript (lines 151-158). Studies of FAPα in additional cohorts of FD patients would confirm our observations. A sentence stating this has been added to the discussion (line 157).
  • Radiolabeled FAPα inhibitors (68Ga-FAPI and 18F-FAPI) imaging by PET/CT maybe an advantageous modality to image FD lesions. Comparative studies using 18F-FAPI and 18F-NaF PET/CT, the current imaging standard, would provide evidence about this application. This is discussed in the manuscript (lines 135-150).
  • Leveraging the specificity of FAPα target peptide sequences and it presence confined to FD lesions may allow the development of peptide-locked pro-drugs that would become active in the local FD microenvironment, allowing the specific treatment of FD lesions while sparing unaffected tissues. For example, a denosumab-like anti-RANKL antibody can be engineered as a FAPα-activable pro-drug, limiting the antiresorptive effects of this drug to FD lesions, as opposed to acting in the whole skeleton. This is discussed in the manuscript (lines 163-185)

In addition to these edits and those in response to reviewer 2, please notice that table 2 (participant demographics) has been updated, as it contained a small error in the number of participants that has now been corrected.

Reviewer 2 Report

Comments and Suggestions for Authors

The authors of a manuscript entitled: “Fibroblast activation protein is expressed by altered osteoprogenitors and associated to disease burden in fibrous dysplasia” present their results of a study focusing on FAP-alpha in the context of fibrous dysplasia (FD). The study is well-designed, and it highlights a critical FAP role as a potential biomarker and a pro-drug activator. While the study is straightforward, the authors must clarify a few points.

1.       The authors describe experiments involving mouse and human cell cultures. It is unclear whether these experiments were a part of the presented study or an already-published work. In particular, the authors provide reference #16 (lines 124-126) in support of the doxycycline-inducible mouse-based model. The reference #16 (Collins, M.T.; Kushner, H.; Reynolds, J.C.; Chebli, C.; Kelly, M.H.; Gupta, A.; Brillante, B.; Leet, A.I.; Riminucci, M.; Robey, 293 PG; et al. An instrument to measure skeletal burden and predict functional outcome in fibrous dysplasia of bone. However, J Bone 294 Miner Res 2005, 20, 219-226, doi:10.1359/JBMR.041111” is not associated with the mouse model.

2.       In the Methods section, the authors indicate (lanes 103-106) that they used FAP human ELISA kit with a sensitivity range of 12 pg/ml to 4000 pg/ml. In Fig 1A, the X axis shows FAP concentration values up to about 12 ng/ml (i.e., 12,000 pg/ml), hence above the applicable range limit. Also, it is not clear if the human-specific ELISA is suitable to detect mouse-derived FAP. The authors must clarify these issues.

3.       Similarly, plasma FAP concentration reaches about 200 ng/ml (i.e., 200,000 pg/ml), a value significantly higher than a functional ELISA upper limit. Consequently, the authors must describe the methods (e.g., serum dilution protocol) they used to ensure the validity of ELISA results.

4.       For the normalized results in Fig 1B, C, and D, the authors should provide more details on the statistical analysis of differences, considering that controls had a value of 1 with no SD after normalization.

5.       If data in Fig 1 are from published studies, the authors must provide specific references in the figure legend.

Author Response

The authors of a manuscript entitled: “Fibroblast activation protein is expressed by altered osteoprogenitors and associated to disease burden in fibrous dysplasia” present their results of a study focusing on FAP-alpha in the context of fibrous dysplasia (FD). The study is well-designed, and it highlights a critical FAP role as a potential biomarker and a pro-drug activator. While the study is straightforward, the authors must clarify a few points.

We appreciate the detailed review of our manuscript, that contributed to increase the quality of our submission. We hope we clarified your questions.

  1. The authors describe experiments involving mouse and human cell cultures. It is unclear whether these experiments were a part of the presented study or an already-published work. In particular, the authors provide reference #16 (lines 124-126) in support of the doxycycline-inducible mouse-based model. The reference #16 (Collins, M.T.; Kushner, H.; Reynolds, J.C.; Chebli, C.; Kelly, M.H.; Gupta, A.; Brillante, B.; Leet, A.I.; Riminucci, M.; Robey, 293 PG; et al. An instrument to measure skeletal burden and predict functional outcome in fibrous dysplasia of bone. However, J Bone 294 Miner Res 2005, 20, 219-226, doi:10.1359/JBMR.041111” is not associated with the mouse model.

We appreciate the reviewer’s thorough review to pick up this mistaken reference. It has now been replaced by the correct citation #15 (Michel, Z.; Raborn, L.N.; Spencer, T.; Pan, K.S.; Martin, D.; Roszko, K.L.; Wang, Y.; Robey, P.G.; Collins, M.T.; Boyce, A.M.; et al. Transcriptomic Signature and Pro-Osteoclastic Secreted Factors of Abnormal Bone-Marrow Stromal Cells in Fibrous Dysplasia. Cells 2024, 13, doi:10.3390/cells13090774.). All the data presented in Figure 1 has been retrieved from publicly available datasets and publications obtained from our literature search and integrated to depict the status of FAP expression in FD. We introduced changes to further clarify that these are previously reported data.

  1. In the Methods section, the authors indicate (lanes 103-106) that they used FAP human ELISA kit with a sensitivity range of 12 pg/ml to 4000 pg/ml. In Fig 1A, the X axis shows FAP concentration values up to about 12 ng/ml (i.e., 12,000 pg/ml), hence above the applicable range limit. Also, it is not clear if the human-specific ELISA is suitable to detect mouse-derived FAP. The authors must clarify these issues.

Thank you for this comment as well, also reflecting a thoughtful review of our manuscript. The mentioned ELISA recommends a dilution of 1:200 for human plasma samples. A sentence has been added to the methods section to reflect this. The mouse BMSCs culture media FAPα concentration showed in Fig 1A is a reproduction of previously reported data from ref #15. The methods corresponding to this measurement can be found in that publication.

  1. Similarly, plasma FAP concentration reaches about 200 ng/ml (i.e., 200,000 pg/ml), a value significantly higher than a functional ELISA upper limit. Consequently, the authors must describe the methods (e.g., serum dilution protocol) they used to ensure the validity of ELISA results.

Please refer to our previous answer (2).

  1. For the normalized results in Fig 1B, C, and D, the authors should provide more details on the statistical analysis of differences, considering that controls had a value of 1 with no SD after normalization.

The relative differences between FD in these graphs were retrieved from publications #15 and #18. Fig 1B corresponds to supplementary table in #15, and Fig 1C corresponds to supplementary tables S4 and S5 in #18, and Fig 1D corresponds to supplementary tables S3 in #18. RNAseq was analyzed for relative genetic expression changes in these publication using DESeq2 algorithm, which provides differential expression data as averages, standard error and multiple measurements-adjusted p values of the experimental groups versus control (=1), but no error data is determined for the control group with this analysis method. To clarify this figure, the following sentence was added in the legend (line 234): “Data is presented as averages and standard deviation in A and average and standard error in B-C.”

  1. If data in Fig 1 are from published studies, the authors must provide specific references in the figure legend.

The references were cited in the legend. We modified the sentence citing these studies to further clarify that this data is adapted from previous publications (line 228): “Data retrieved and adapted from published studies and genetic expression datasets [15,19]“

In addition to these edits and those in response to reviewer 1, please notice that table 2 (participant demographics) has been updated, as it contained a small error in the number of participants that has now been corrected.

Round 2

Reviewer 1 Report

Comments and Suggestions for Authors

Dear Authors,

Thank you very much for submitting your work.

Reviewer 2 Report

Comments and Suggestions for Authors

The authors addressed all original critiques adequately.